# The Effect of Strengthened Physical Education on Academic Achievements in High School Students: A Quasi-Experiment in China

**DOI:** 10.3390/ijerph16234688

**Published:** 2019-11-25

**Authors:** Yunting Zhang, Xiaochen Ma, Jin Zhao, Hong Shen, Fan Jiang

**Affiliations:** 1Child Health Advocacy Institute, Shanghai Children’s Medical Center, Shanghai Jiao Tong University School of Medicine, Shanghai 200127, China; edwinazhang@sjtu.edu.cn (Y.Z.); zhaojin@scmc.com.cn (J.Z.); 2China Center for Health Development Studies, Peking University, Beijing 100191, China; xma@hsc.pku.edu.cn; 3The Affiliated High School of University of Shanghai for Science and Technology, Shanghai 200093, China; xzshenhong@126.com; 4Department of Developmental and Behavioral Pediatrics, Shanghai Children’s Medical Center, Shanghai Jiao Tong University School of Medicine, Shanghai 200127, China; 5Ministry of Education-Shanghai Key Laboratory of Children’s Environmental Health, Shanghai 200092, China

**Keywords:** physical education, academic achievements, high school students, China

## Abstract

Objectives: Evidence of school-based physical activity (PA) on academic performance in children and adolescents was inconsistent, especially in high school students who face a high academic burden. In this study, we tested the efficacy of a strengthened physical education (PE) program on academic outcomes in Shanghai. Methods: A quasi-experimental design was conducted to investigate the effect of strengthened PE on academic scores by calculating the grade-cohort difference before and after the intervention. PE curriculum switched from traditional short duration (40 minutes) general fitness training to long duration (90 minutes) specialized sports (e.g., football, aerobics). A total of 460 high school students (236 pre-intervention and 224 post-intervention) were enrolled in grade 10 and followed for two and three semesters. The academic outcome was assessed by district-standardized test scores. A difference-in-difference approach was employed. Results: After two semesters, the standardized Chinese language scores and English language scores for the post intervention group were increased by 0.61 SD (95% confidence interval (CI): 0.44, 0.78, *p* < 0.001) and 0.28 SD (95% CI: 0.09, 0.47, *p* = 0.01). However, the standardized math scores for the post intervention group were decreased in girls. After three semesters, standardized Chinese language scores for the post intervention group were increased by 0.27 SD (95% CI: 0.06, 0.48, *p* = 0.01). Math scores and English language scores decreased by 0.18 SD (95% CI: −0.36, −0.01, *p* = 0.04) and 0.23 SD (95% CI: −0.38, −0.09, *p* = 0.00), respectively. Conclusion: A school-based physical education program had mixed effects on academic scores in high school students.

## 1. Introduction 

The growing epidemic of obesity in children and adolescents has become a global health crisis, for it is associated with type 2 diabetes, specific types of cancer, and cardiovascular disease among adolescents [1]. One well-documented key factor is inadequate physical activity (PA) [2]. A recent national survey in China found that less than 30% of school-aged children met the criteria of at least 1 h/day of moderate-to-vigorous physical activity (MVPA) [3]. School based PA, especially curricular physical education (PE), has an unparalleled opportunity to promote children’s health by engaging them in regular physical activity [4], while at the same time, advocates have long argued for the necessity of school-affiliated PA [4,5]. Critics, mostly parents and educators, however have worried that the increasing PA would come at the expense of decreasing the time allocated to more valued courses, like language or math [6]. Moreover, the opposition becomes stronger when students reach higher grade levels in high school and face college admission decisions [7]. As a result, we have witnessed a global trend of decreasing PA given the increased academic burden during adolescence [8]. 

On the contrary to the worrisome, studies have suggested that the time spent in PA would benefit cognition among children and adolescents and would not endanger their academic performance [9,10,11]. Evidence suggested that physical activity can promote cognitive function through structural and functional changes in the brain, including improved brain plasticity, stronger connections between brain regions, and even increased volume of grey matter and white matter [12,13,14,15]. However, two systematic reviews found that the effects of PA interventions on academic performance are mixed. Some studies that have investigated additional or enhanced PE did not show positive results with increased academic performance [16,17]. Therefore, one of the urgent needs is to acquire more evidence of what kind of PA intervention in a school environment would not endanger, or even benefit, their school performance.

Strengthened PE lessons usually take the form of increasing time or enhancing PA intensities. Increasing PE sessions alone does not necessarily guarantee cognitive benefits [18,19]. While the traditional PE curriculum in China is designed to promote endurance, strength, flexibility exercises, and circuit training for cardiovascular health, studies are emerging focusing on the cognitive enrichment of exercises, which involve motor learning, interactions with the environment, and group cooperation requiring mental operations [20,21,22]. Structured and complex sports are likely to improve several cognitive processes such as attention, concentration, and inhibitory functions, which play a crucial role in the achievement of academic goals and life skills that are essential to positive youth development [23,24]. 

The aim of our study was to examine the effect of a strengthened PE pilot program consisting of specialized sports training on students’ academic performance in the high school context. The strengthened PE session focused on improvement of both the length and content. Specifically, we tested whether the effects were different across different academic subjects and whether the effects were different over time.

## 2. Method 

### 2.1. Setting 

Starting from January 2015, one high school in Yangpu district of Shanghai municipal city was selected to participate in the PE pilot program supported by the Shanghai Educational Bureau. The goal of this program was to improve the interest and participation in PA at the school, as well as involving cognitive enrichment with a shift from a 40-minute traditional cardiovascular fitness oriented curriculum to a 90-minute specialized sports training oriented curriculum. Prior to the program, the PE curriculum included four 40-minute sessions per week: three regular PE sessions and one free activity session. In the regular PE session, students were asked to participate in general fitness training such as running, standing long jump, throwing shot, pull-up, sit-up and etc. In the free activity session, no specific training protocol was used. Without changing the total sessions of PE courses, the pilot school re-modeled the curriculum that combined two 40-minute sessions and the 10-minute break in between into one 90-minute session. The content of PE session was also changed from general fitness training to specialized sports training. Specialized sports training—coached by PE teachers with relevant expertise—included football, volleyball, badminton, table tennis, tennis, and aerobics. Students got a chance of choosing the sports of their own interest at the beginning and took the training of this special sports during all three years in their high school. Activities in the new PE sessions included basic skills coaching for the specialized sports, fitness training required, and a competitive game to engage all students with real practice. Therefore, the program changed the PE curriculum from four 40-minute general fitness training or free activity sessions (160 minutes per week) to two 90-minute specialized sports training sessions (180 minutes per week). By design, increasing the session duration from 40 minutes to 90 minutes had two benefits: first, it would presumably increase the proportion of MVPA in a typical PE lesson by avoiding a duplicated warm up and cool down during the starting and ending; second, a 90-minute session made it less likely to be occupied by other academic courses, which typically consisted of 40-minute sessions. Also, changing from fitness training to specialized sports increased the engagement of students’ participation of sports owing to personal interest and more chances of group activities. The PE curriculum before and after the intervention was recorded by the school administration database, information was also available on which specialized sports was chosen by each individual participants.

The study was approved by the institutional review board of the Shanghai Children’s Medical Center (SCMC), Shanghai Jiao Tong University (SCMCIRB-K2018021).

### 2.2. Study Design and Sampling 

Two consecutive grade-cohorts were established before (*pre-intervention* cohort) and immediately after (*post-intervention* cohort) the PE pilot program took place. The *pre-intervention* cohort was used as the control group for the intervention cohort. Both cohorts were followed for two and three semesters after the first semester at grade 10 (baseline). Figure 1 presents the study flow chart: the September 2013 *pre-intervention* cohort enrolled 236 students in grade 10 (baseline). They were taught traditional PE sessions during the whole study period (baseline and two follow-ups). The September 2014 *post-intervention* cohort enrolled 224 students in grade 10 (baseline). They were taught the traditional PE sessions in their first semester (baseline), but this cohort was followed by the strengthened PE sessions during the two follow-ups. The follow-up rate was 93% for two semesters and 91% for three semesters. 

### 2.3. Outcome Measures and Covariates 

The outcome was the difference in change in academic achievement between the *pre-intervention* cohort and the *post-intervention* cohort over the course of two and three semesters. The Educational Bureau of Yangpu District administrated standardized tests of Chinese language, English language, and math to all high schools in its district at each semester. For each student, their raw scores at each semester were converted to a standardized score by subtracting with the district mean and divided by the district standard deviation. This standardized score can be interpreted as the relative position of each student in the whole distribution of Yangpu district. Therefore, the standardized score allowed us to compare academic performance among students in different grade-cohorts. Hence, we have three outcome variables measured by the standardized score of Chinese language, English language, and math. Data links with the *Yangpu District Educational Bureau School Information Management System* provided the results of the assessments in Chinese language, English language, and math on both the individual level and district level. 

A series of potential confounders were adjusted: age, gender, obesity (height and weight from Shanghai municipal physical fitness evaluation system, and weight status calculated based on World Health Organization norm) [25], and family wealth measured by the real estate price of the subject’s current neighborhood. Two teaching quality indicators were also added: years of teaching experience [26] and credential rank (measured from 1 to 10 based on an annual assessment of local educational administrators, higher rank indicates better teaching quality) [27]. 

### 2.4. Statistical Analysis 

Change in academic achievements was calculated by subtracting the standardized scores at the second and third semester follow-up from the baseline score collected in the first semester at grade 10. Changes in academic achievements over time were then compared between students in *prior-* and *post-intervention* cohorts (difference-in-difference method, DID).

Two regression models were used to explore the effect of strengthened PE on the academic achievements of students adjusted for different sets of confounders. Model 1 (minimally adjusted model) was adjusted for the baseline standardized score. Model 2 (fully adjusted model) was adjusted for the baseline score and the rest of the confounders. The fully adjusted model was repeated for boys and girls separately.

All statistical analyses were performed using STATA 14.1 (StataCorp, College State, TX, USA).

## 3. Results 

### 3.1. Characteristics of Study Participants

Table 1 compares the baseline characteristics between the *pre-intervention* cohort and the *post-intervention* cohort. There was no significant difference in baseline student and family characteristics between the two cohorts. Teaching experience and credential rank of Chinese language, English language, and math teachers were not always comparable, but no consistent pattern was found.

### 3.2. Associations with Academic Performance in Two-Semester Follow-Up 

Figure 2 plots the effect size of the difference-in-difference (DID) analysis adjusted for the baseline score only (Model 1) and the full set of covariates (Model 2). Similar patterns assessed at two semesters were found in both models, although Model 2 had slightly larger effect sizes. In Model 2, students taught strengthened PE had higher Chinese (0.61 standard deviation (SD), 95% confidence interval (CI): 0.44 to 0.78, *p* < 0.001) and English scores (0.28 SD, 95% CI: 0.09 to 0.47, *p* = 0.01), but lower math scores (−0.27 SD, 95% CI: −0.42 to −0.11, *p* = 0.00), than those with the traditional PE session. For Chinese language and English language scores, direction and effect sizes were similar in both genders. For math tests, however, a gender effect was found: strengthened PE was associated with lower scores among girls (−0.36 SD, 95% CI: −0.57 to −0.16, *p* < 0.001), but no significant association was found among boys (Table 2).

### 3.3. Associations with Academic Performance in Three-Semester Follow-Up 

Assessed at three semesters, similar results were found in both models. In the fully adjusted model, the positive association between strengthened PE session and Chinese scores persisted, although with a mitigated effect size of (0.27 SD, 95% CI: 0.06 to 0.48, *p* = 0.01). The positive association found in English scores became negative (−0.23 SD, 95% CI: −0.38 to −0.09.47, *p* = 0.00). The negative association found in math scores still persisted (−0.18 SD, 95% CI: −0.36 to −0.01, *p* = 0.04). Similar gender differences were found at the three-semester follow-up (Table 2).

## 4. Discussion

### 4.1. Main Findings and Study Implications

PE is the cornerstone of the school-based PA program [28]. For Chinese high school-aged adolescents, the amount of time allocated to physical activities is under the recommended level [3]. This study evaluated, through a quasi-experimental design, a strengthened physical education pilot in one of the world most academically competitive high school settings, the Shanghai metropolitan area in China [29]. The experience and lessons from this pilot program will help re-model the physical education curriculum across all high schools in Shanghai. Taking advantage of a quasi-experimental design, we were able to investigate the effect of a strengthened PE program on academic performance among high school students across different academic subjects and over time. To our knowledge, this is the first study to test the effect of a strengthened PE program in a high school setting from East Asian countries. 

Several key findings are highlighted. First, participating in strengthened PE for two semesters had a significantly positive effect on overall academic performance of high school students, especially in Chinese language and English language scores. Studies conducted in high school students are rare and inconsistent [30,31]. In this study, we were able to control important confounding variables such as baseline attainment and teacher quality. This helped to argue against other alternative explanations on the change of academic performance. The positive findings of the present study were also in agreement with a broader range of studies exploring the association between physical activity and academic performance in children and adolescents [20,32,33,34,35,36]. However, comparing the effect size in our study to others was challenging owing to differences in the form and intensity of school-based PA [37], measurements of academic performance [20,35], study population, and contexts. 

Second, the effects of strengthened PE were different across academic subjects and gender: language scores for Chinese and English increased, but scores in math decreased. Although one recently published systematic review found strong evidence for the beneficial effects of PA on math performance [17], all five studies included in this review were conducted among elementary school students. One possible explanation for the nonpositive negative results in our high school setting was that cognitive skills required for math learning are different across stages of schooling [38]. In Chinese high schools, mathematics lessons are already very sophisticated and require advanced logical reasoning ability, which is less likely to be improved by incremental school-based physical activity [32]. This subject-specific effect in our finding was in line with that in laboratory studies [14,39]. Physical fitness is associated with greater bilateral hippocampal volumes and superior relational memory task performance, which was beneficial to language abilities, but not math skills [40]. Interestingly, we found the negative effect on math score occurred primarily among girls. This gender differential effect has also been discovered in other studies [41,42]. Two possible pathways might underlie this: first, boys and girls may have chosen different types of sports. Evidence consistently supports that not all forms of exercise influence cognition equally [43]. In our sample, we found that, compared with boys, girls are more likely to choose individual sports (badminton, table tennis, tennis) over group sports (football, volleyball, aerobics). We also found a different effect on math scores across these two sport types (Appendix A). Second, the mechanism of how PA has an impact on academic performance may be different across gender [41,42]. 

Third, the differential results over time in which the positive effects at two semesters’ follow-up were attenuated at three semesters paralleled the concerns of programme sustainability caused by premature discontinuation in the implementation science literature [44]. On the basis of our interviews with school teachers, we believed it was because the pilot school did not continue strictly with the intervention plan in the third year of high school because of the pressure of preparing for the national college entrance examination [45]. Actually, out of the two 90-minute PE sessions per week, one of them was “downgraded” to a 40-minute optional activity session in which the intensity of the activity was lighter. In many cases, this session was occupied by other academic courses. This finding, in combination with previous studies in other settings, shows the challenges of translating findings from clinical trials into policy that yields population-level benefits, suggesting that the success of the translation requires pragmatic, localized application of evidence-based intervention strategies into a real-world setting [46]. In our case, the real-world barriers apparently dominated the rationale for strengthened PE when the students faced intense academic pressure. 

### 4.2. Limitations 

First, the quasi-experimental design, by nature, was inferior to a randomized controlled trial. However, we used the difference-in-difference approach to mimic an experimental research study design. Also, this design enables us to evaluate this PE reform in a real world situation, which, to some extent, will be more applicable to implementation. Second, in this study, we did not have an objective measurement of physical activity, and thus it is not clear if there was a significant difference in duration or intensity of physical activity, which limited our ability to understand the underlying mediators. Even if we are not able to tease out whether the beneficial effects in language skills were driven from the level of physical activity per se or the content provided in PE class, the nonnegative evidence gathered in this study was useful in advocacy with the policy makers to establish programs in a large scale. Third, our estimates of the effect of strengthened PE on academic achievement in high school students applied to adolescents facing high academic burden and lack of physical activities—a population of considerable interest for health and education policies. However, the generalizability of our results should be limited to the context of China or a similar East Asian setting with similar contextual factors. 

## 5. Conclusions

Our study found a positive effect of strengthened PE with a specialized sports training focus on Chinese and English language scores of high school students, even in a population with the heaviest academic burden. However, the discontinuation of the intervention, and thus the non-effect in the longer term, revealed the challenges on programme sustainability. Nevertheless, the findings in our study provide new evidence applicable to school-based PE curriculum design with more cognitively engaging forms of exercise in adolescents. 

## Figures and Tables

**Figure 1 ijerph-16-04688-f001:**
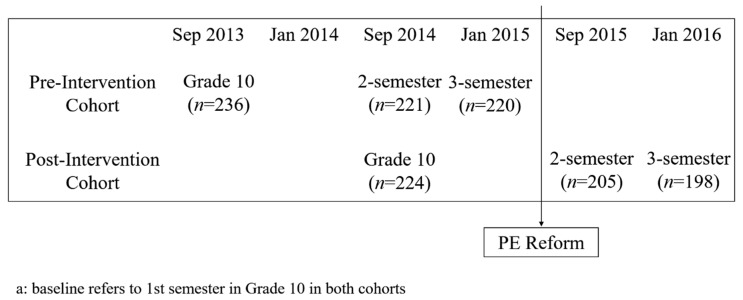
Flow chart of the quasi-experiment on physical education (PE) reform.

**Figure 2 ijerph-16-04688-f002:**
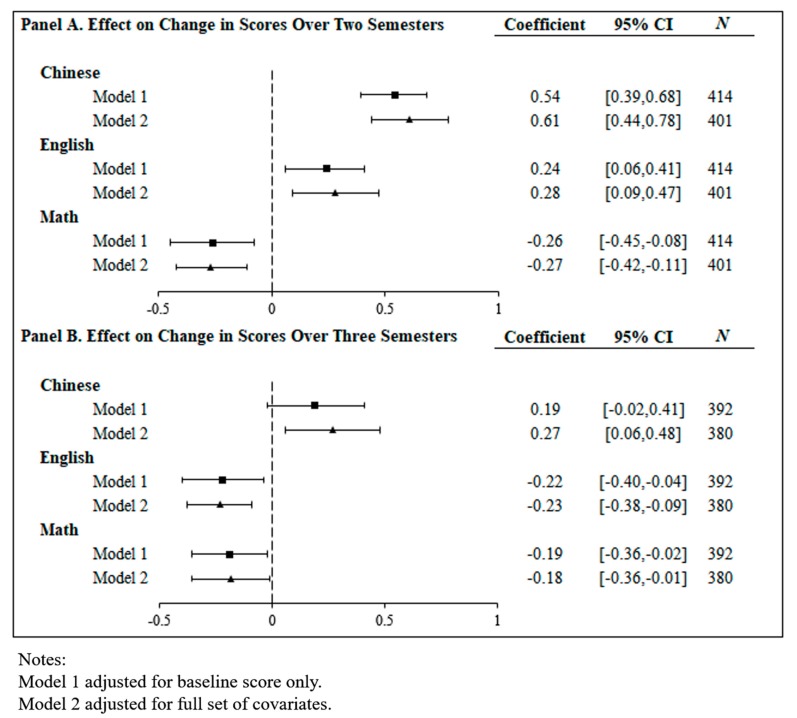
Effect size of difference-in-difference (DID) analysis adjusted for 192 baseline score only (Model 1) and the full set of covariates (Model 2). CI, confidence interval.

**Table 1 ijerph-16-04688-t001:** Baseline characteristics between the pre-intervention cohort and the post-intervention cohort.

Characteristics		Pre-Intervention Cohort	Post-Intervention Cohort	*p*-Value
Students				
	Total participants (n);	236	224	
	Age (years)	15.30 (0.48);	15.38 (0.51)	0.06
	Female (%)	58.90	55.36	0.44
	Weight			0.35
	Normal (%);	70.23	66.00	
	Overweight or obese (%)	29.80	34.00	
	House price (10,000 RMB/sq meter)	6.15 (1.42)	5.98 (2.05)	0.30
Teachers				
Grade 11 Teachers				
Chinese Teachers				
	Total participants (n)	8	8	
	Teaching experience (years)	24.75 (7.30)	18.75	0.05
	Credential rank	7.75 (1.58)	9.25 (0.89)	0.03
English Teachers				
	Total participants (n)	8	8	
	Teaching experience (Years)	15.75 (2.96)	23.50 (9.15)	0.04
	Credential rank	9.75 (1.04)	8.75 (2.31)	0.28
Math Teachers				
	Total participants (n)	8	8	
	Teaching experience (Years)	17.50 (6.54)	19.25 (4.74)	0.55
	Credential rank	9.13 (2.16)	8.75 (2.31)	0.74
Grade 12 Teachers				
Chinese Teachers				
	Total participants (n)	8	8	
	Teaching experience (Years)	21.50 (8.01)	20.25 (6.16)	0.73
	Credential rank	8.50 (2.20)	8.25 (1.90)	0.81
English Teachers				
	Total participants (n)	8	8	
	Teaching experience (Years)	18.13 (8.00)	17.13 (4.76)	0.77
	Credential rank	9.75 (1.75)	9.50 (1.41)	0.76
Math Teachers				
	Total participants (n)	8	8	
	Teaching experience (Years)	23.88 (10.34)	16.38 (4.10)	0.08
	Credential rank	8.25 (2.12)	9.13 (1.46)	0.35

**Table 2 ijerph-16-04688-t002:** Associations of strengthened physical education with academic performance among boys and girls in two- and three-semester follow-up. CI, confidence interval.

**Panel A. Effect on Change in Scores Over Two Semesters**					
****	****	**Coefficient**	**95% CI**	***p*-value**	**Sample Size**
Boys					
	Chinese	0.58	[0.30, 0.86]	<0.001	167
	English	0.33	[0.12, 0.53]	<0.001	167
	Maths	−0.13	[−0.31, 0.04]	0.12	167
Girls					
	Chinese	0.63	[0.39, 0.86]	0	234
	English	0.18	[0.03, 0.34]	0.02	234
	Maths	−0.36	[−0.57, −0.16]	0	234
**Panel B. Effect on Change in Scores Over Three Semesters**					
****	****	**Coefficient**	**95% CI**	***p*-value**	**Sample Size**
Boys					
	Chinese	0.27	[0.04, 0.51]	0.03	158
	English	−0.21	[−0.40, −0.02]	0.03	158
	Maths	−0.03	[−0.20, 0.14]	0.69	158
Girls					
	Chinese	0.25	[−0.01, 0.51]	0.06	222
	English	−0.26	[−0.42, −0.09]	0	222
	Maths	−0.29	[−0.49, −0.08]	0.01	222

Notes: fully adjusted model was repeated for boys and girls separately.

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
