# Peer review of "The Effect of Strengthened Physical Education on Academic Achievements in High School Students: A Quasi-Experiment in China"

_ijerph, 2019, doi:10.3390/ijerph16234688_

Round 1

Reviewer 1 Report

Please find my detailed review in the attached PDF file.

Author Response

First, given the samples in our study are all from one single school in Yangpu District, the reviewer asked about how representative is our sampled school. In two ways, our sample school is representative and of good policy relevance to our research questions. At the district level, the whole program of this strengthened PE curriculum is for all the schools in Shanghai. And 38 high schools from 17 districts were chosen as pilot schools. The pilot schools are all above average level in terms of teaching quality to ensure the implementation of program. Among all the 17 districts in Shanghai, we chose Yangpu District, which is one of the most competitive districts in terms of academic achievement, because we want to test our hypothesis in the most academic burdened students. At the school level, the reason for selecting this school in Yangpu district for our study was because it is the one of key schools of Yangpu district, which in many ways (students quality, teacher quality, school infrastructure) have good representativeness for this district. Please find a link with the school introduction as following: http://fwpt.yp.edu.sh.cn/shlgdxfszx/info/1017/1407.htm

Second, the reviewer expected us to elaborate the choice for two specific control variable’s: family wealth, and teacher quality.

We used family wealth rather than family income as control variable for two reasons: a), this is a retrospective study, students in this study were all graduated, and therefore it is very difficult to obtain their family income; b) The housing price in each community is open source data, which is more accurate. And in Shanghai, the housing price is extremely high to account for a majority proportion in family asset.

Regarding the teaching quality, since the dependent variable in our model is English, Chinese and Math scores, and the teaching quality is important confounder on changes of academic performance. However, those quality variables were not controlled in previous studies. Therefore we include the teaching quality of these three subjects in the covariates to control for its effect on the academic scores. And for the expertise of PE teachers, we do admit that they could have influence on students’ participation. However, since the curriculum has been shifting from cardiovascular training to specialized sports, and they were not evaluated officially on their coaching skills of these sports activities. Therefore we did not include this in the model.

Third, the reviewer asked whether there have been studies investigating the effect of increase PE in the rest of the world. Yes, this is a topic of growing interest and there are quite a lot of studies investigating the effect of increase PE in the curriculum. But as we indicated in our manuscript, most of the existing studies are in younger children and in western countries. Therefore, to our best knowledge, our work is the first study to test the effect of strengthened PE program in high school setting from East Asian countries.

Fourth, the reviewer suggested that the choice of sport might help understand the gender differential effect on academic performance, particularly on math score. We highly appreciate this valuable suggestion. We conducted additional analysis by adding types of sports in the regression model. Altogether there are six types of specialized sports curriculum, which are football, volleyball, badminton, table tennis, tennis and aerobics. Because our sample does not have sufficient statistical power to test the difference across six sports. We categorized them into individual sports (badminton, table tennis, tennis) and group sports (football, volleyball, aerobics). We found a clear gender difference in the choice of sports: girls are more likely to choose individual sports (Boys: 74% group vs 26% individual; Girls: 34% group vs 66% individual). Adding this new variable into our regression model, we found different effect on Math scores across this two sport type. In the second semester, the effect is -0.18 for group sports and -0.37 for individual sports. In the third semester, the effect is -0.13 for group sports and -0.25 for individual sports. No significant difference was found in effect on Chinese and English. This supports our speculation that the gender heterogeneous effect on math score is because the girls choose different sport type than boys. We have also added this part of the analysis in the appendix file. We have also expanded our discussion section to include this point in the third paragraph of the discussion.

First, boys and girls may have chosen different types of sports. Evidence consistently supports that not all forms of exercise influence cognition equally44 45. In our sample, we found that compared to boys, girls are more likely to choose individual sports (badminton, table tennis, tennis) over group sports (football, volleyball, aerobics). We also found different effect on Math scores across this two sport type (Appendix Table 1).

Last but not least. We edited our manuscript for the typos as the reviewer pointed out.

Reviewer 2 Report

My biggest general concerns relate to:

The literature study that is rather limited and focuses in part on the problem of too little exercise, while this research focuses on examining the effects of physical activity on the learning outcomes of higher education students. The literature study pays only limited attention to what scientific evidence already exists about the role that physical activity can play in facilitating learning. In other words, what people already know about the effect of physical activity on the functioning of the brain. While this is now the basis of the research design. In the literature section, the authors already state that they will work with a strengthened program in physical education. The way in which this is compiled is and in what way it is fed by evidence-based information is completely missing. When I consider the methodological section, I believe that few scientific foundations are cited to support the added value of the intervention. The way in which the intervention is currently described in this section suggests that the authors had no say in the curriculum change. If this is the case, it is recommended that the authors rename their studies. A second concern within the methodological section is the lack of measurements that register the physical activity of the target group as well as the motivation to participate with the participating students, while that is supposed to be an added value of curriculum reform. The authors state in the method section that the curriculum change entails an increase in engagement with physical activity, while this is not further investigated in the research design. The above concerns have a major impact on the results, the discussion and the conclusion section because the original focus of the research is lost and it is therefore difficult for the authors to make a critical analysis. Finally, the conclusion section of this work contains a number of very far-reaching generalizations, whereby too little account is taken of the limited quality and unchecked assumptions about the impact of the reform implemented.      

Author Response

First, the reviewer suggested that more literature on physical activity on the functioning of brain is needed. In the introduction part, we have added some literature investigating effect of physical activity on the functioning of the brain in the second paragraph of the introduction as following:

Evidence suggested that physical activity can promote cognitive function through structural and functional changes in the brain, including improved brain plasticity, stronger connections between brain regions and even increased volume of grey matter and white matter

Second, the reviewer asked for more evidence-based information on the design of our intervention. We have added a paragraph addressing the evidence related to the design of this intervention in the third paragraph of the introduction as following:

Strengthened PE lessons usually take form of increasing time or enhancing PA intensities. Increasing PE sessions alone does not necessarily guarantee cognitive benefits. While the traditional PE curriculum in China was designed to promote endurance, strength, flexibility exercises and circuit training for cardiovascular health, studies are emerging focusing on the cognitive enrichment of exercises, which involve motor learning, interactions with the environment and group cooperation require mental operations. Structured and complex sports are likely to improve several cognitive processes such as attention, concentration and inhibitory functions, which plays a crucial role in the achievement of academic goals and life skills that are essential to positive youth development.

We would also like to mention that, one of our co-authors Mr. Shen Hong, is the principle of this pilot school. He himself is a PE teacher and he was in charge of the design and overseen the implementation of this intervention.

Third, the reviewer was concerning that there was lack of measure on the engagement of physical activity. We appreciate the reviewer for pointing this out. For the engagement of physical activity, it was not measured in the two cohorts and we have already admitted this in our limitation section in our original manuscript.

In the methodology part, we added the clarification of how students were assigned to different specialized sports groups. The students chose the sports that they are interested in and after the decision was made, they took the training of this special sports during all three years in their high school. Athletic skills for this specific sports were taught step by step thought the three years.

Finally, the reviewer suggested that given our study design and evidence presented it was over-generalizing to interpret our results to the effect of physical activity, rather it should be narrowed to the effect of a strengthened PE program. Again, we highly appreciated this valuable comment. Indeed, the goal of our study is to test the effect of a PE intervention program on students’ academic score in a real world situation and to provide new evidence in this area. We have removed the part related to physical activity, which was not a key variable of interest of this study, and rewrote the concluding sentence as the following:

Our study found positive effect of strengthened PE with specialized sports training focus on Chinese and English language scores of high school students, even in a population with the heaviest academic burden. However, the discontinuation of the intervention therefore non-effect in the longer term revealed the challenges on programme sustainability. Nevertheless, the findings in our study provide new evidence applicable to school-based PE curriculum design with more cognitively engaging forms of exercise in adolescents.

Reviewer 3 Report

General comments.
The work presents two intervention groups in two moments that could affect the interpretation of the results, as conditions change.

On the other hand, no positive results are obtained with the intervention, so it is difficult to transfer for other researchers to replicate the study.

Introduction

Line 44. Is necessary to explain the sentence “achievement were mixed”.

Line 46. To explain why the “cognition is critical for adult health”. Be more precise…

Line 49. To correct the space between words.

Line 53. To correct the space between words.

Line 54 to 62. This sentences are unnecessary. We deed delete this sentences.

This can be added in the discussion, but not here.

You should use that space to justify the study based on existing evidence. It cannot justify with the authors' own opinion.

Line 62 to 65. In this sentences you must explain what the “PE pilot program” consists of.

Methods.

Line 69. The acronym “PE” has been described in introduction. No repeat.

Line 94 to 102. The first cohort was the control group? You need to explain better.

Moreover, the figure 1 isn`t of good quality. It must be replaced.

Results.

Is necessary show the Figure 2 in a better quility.

The table 2 is showed with de results in the vertical line. I suggest that they be presented in the horizontal.

Discussion.

Line 207-207. What is meant is not well understood.

Line 213. How can this statement be made (intense academic pressure) if it has not been evaluated?

Author Response

We appreciated the comments from reviewer. Yes we did not find consistent positive result in our study, which is similar to other studies in this area. However we still believe that our study adds to the current literature with the following value:

Quasi-experimental design exploring the effect of a scalable strengthened physical education program on academic performance. Conducted in the highly academic burden school setting. Mixed results found across different academic subjects and over time with a significant improvement in language related subjects.

Introduction

Line 44. Is necessary to explain the sentence “achievement were mixed”.

A systematic review suggested that the association between PA participation and academic performance was not consistent. The detailed statements are as following: Overall, the intervention studies that have investigated additional or enhanced PE did not show positive results, with only two of six finding any positive effect of a PE program on achievement scores.

We have added further explanation in the manuscript in the second paragraph of introduction.

Line 46. To explain why the “cognition is critical for adult health”. Be more precise…

Cognitive processes such as attention, concentration and inhibitory functions, which plays a crucial role in the achievement of academic goals and life skills that are essential to positive youth development. We have revised this in the last sentence of the third paragraph of the introduction part.

Line 49. To correct the space between words.

Thank you, we have made the correction.

Line 53. To correct the space between words.

Thank you, we have made the correction.

Line 54 to 62. This sentences are unnecessary. We deed delete this sentences.

This can be added in the discussion, but not here.

You should use that space to justify the study based on existing evidence. It cannot justify with the authors' own opinion.

Thank you! We totally agree with this comment. And we have revised the introduction according to this suggestion. We added the rationale of this intervention in the third paragraph of the introduction.

Line 62 to 65. In this sentences you must explain what the “PE pilot program” consists of.

Thank you! We have added that the PE pilot program was consisted with specialized sports training in this sentence. Detailed explanation would be found in the following methodology-setting section.

Methods.

Line 69. The acronym “PE” has been described in introduction. No repeat.

Thank you, we have made the correction.

Line 94 to 102. The first cohort was the control group? You need to explain better.

Moreover, the figure 1 isn`t of good quality. It must be replaced.

Thank you, we have made further explanation in this section. And we have also re-plotted figure 1.

Results.

Is necessary show the Figure 2 in a better quility.

The table 2 is showed with de results in the vertical line. I suggest that they be presented in the horizontal.

Thank you! We have made the revision of table 2.

Discussion.

Line 207-207. What is meant is not well understood.

Usually when students enter the third year of high school. Their main focus is the national college entrance examination. From our interview with the school teachers, the PE lesson were occupied by other academic subjects in the last year. We also revised the sentence in the manuscript to make it easier to understand.

Line 213. How can this statement be made (intense academic pressure) if it has not been evaluated?

Thanks to this good point. We realize that the Chinese context might be difficult for international readers to understand. To evaluate whether academic pressure increase in grade 12 is beyond the scope of this study. The college entrance examination is so important that they are commonly dubbed "once in a lifetime" or a "one-test-to-determine-a-life" for most high school graduates. Given the increasing school burden in the last year of high school is a common phenomenon in China, we added a citation in the sentence in the fourth paragraph of Discussion section.

Reviewer 4 Report

This is an important manuscript that will add to understanding of the academic impact of PA.  The author(s) should include more evidence in the literature review of other research examining PA and academics.  There is an increasing body of literature that would be important to cite.  Good explanation for the rationale for the study.  More details on the actual intervention are needed.  What did specialized sports training include?  How was this different than the previous PE classes?  Expand on the intervention.  Also, what were the hypotheses for the study?  Did the authors expect to find differences in any of the scores?  Was the time frame sufficient to notice a different in scores and would they be specific to only the PE intervention?  What else might have influenced differences? Also, what would be of importance to include in future interventions?  The authors(s) note it is a challenge to implement interventions in real world settings, but would might assist in terms of recommendations that could be given for future such practice implications?  

Author Response

This is an important manuscript that will add to understanding of the academic impact of PA. 

Thank you very much for your comment.

The author(s) should include more evidence in the literature review of other research examining PA and academics.  There is an increasing body of literature that would be important to cite.  Good explanation for the rationale for the study. 

Thank you very much! We have revised the introduction part and added more literature in this part, as well as the rationale for this study.

More details on the actual intervention are needed.  What did specialized sports training include?  How was this different than the previous PE classes?  Expand on the intervention. 

Thank you! After the intervention, PE lessons were centered on the specialized sports that the students chose, which included the basic skills for the specialized sports, the fitness training required and a competitive game to engage all students with real practice. In the previous regular PE session, students were asked to participate general fitness training such as running, standing long jump, throwing shot, pull-up, sit-up and etc. We have added the details in the methodology-setting part.

Also, what were the hypotheses for the study?  Did the authors expect to find differences in any of the scores? 

We believed our hypothesis was listed in the end of Introduction section. And we expected to find differences on students’ academic scores with this strengthened PE curriculum.

Was the time frame sufficient to notice a different in scores and would they be specific to only the PE intervention? 

Yes we believe that the time frame was sufficient to notice a difference in scores. Usually the school based PA intervention study take time from 6 months to 2 years, as the previous studies we reviewed in the introduction and discussion sections and also studies included in several systematic reviews as following:

Mura G, Vellante M, Egidio Nardi A, Machado S, Giovanni Carta M. Effects of school-based physical activity interventions on cognition and academic achievement: a systematic review. CNS & Neurological Disorders - Drug Targets (Formerly Current Drug Targets - CNS & Neurological Disorders) 2015. Singh AS, Saliasi E, van den Berg V, et al. Effects of physical activity interventions on cognitive and academic performance in children and adolescents: a novel combination of a systematic review and recommendations from an expert panel. Br J Sports Med 2018

In our study, the two cohorts are in the same school with just one year gap for school entry. Therefore we believe the PE curriculum is the only factor that differs between those two cohorts. We also adjusted the quality and experience of teachers to control for their influence on students’ academic performance.

What else might have influenced differences?

We believe that students’ academic baseline, the teachers’ quality and after school curricular would have greater influence on their academic scores. And also gender difference will also be seen in the academic scores. Therefore we include the teaching quality and expertise of the Chinese, maths and English teachers in the covariates to control for its effect on the academic scores. And we also did the sensitivity analysis across gender.

Also, what would be of importance to include in future interventions?  The authors(s) note it is a challenge to implement interventions in real world settings, but would might assist in terms of recommendations that could be given for future such practice implications?  

We would suggest the inclusion of the measurement of physical activity in the future interventions to test the exact change for physical activity participation individually. For recommendations that could be given for future practice, given the real-world barriers of a school-based PE program from the increasing academic burden, we would suggest that school-based intervention should consider strategies to engage key stakeholders besides educators, especially parents.

Reviewer 5 Report

Very well developed research.  However, the authors should elaborate on the background of the study developing the introduction part with special reference to the cognitive effects of PA in school children. Besides, a few sentences on the importance of this study should also be inserted in the text. The article by Singh et al. (2018) would inspire you more to develop this part. The conclusion part needs to be improved accordingly. I would strongly recommend the authors continue studying on the same subject using different research methods as we have blurred even contradictory findings in this study. 

Author Response

Thank you very much for your valuable suggestion. We have revised the introduction and conclusion part and also reference the article in the manuscript. This article inspired us with new ideas of continuing studying on this subject with more rigid design.